# Cold Drawing of AISI 321 Stainless Steel Thin-Walled Seamless Tubes on a Floating Plug

**DOI:** 10.3390/ma16165684

**Published:** 2023-08-18

**Authors:** Krzysztof Żaba, Tomasz Trzepieciński

**Affiliations:** 1Department of Metal Working and Physical Metallurgy of Non-Ferrous Metals, Faculty of Non-Ferrous Metals, AGH—University of Science and Technology, al. Adama Mickiewicza 30, 30-059 Cracow, Poland; 2Department of Manufacturing Processes and Production Engineering, Faculty of Mechanical Engineering and Aeronautics, Rzeszow University of Technology, al. Powst. Warszawy 8, 35-959 Rzeszów, Poland

**Keywords:** cold tube drawing, floating plug, AISI 321 stainless steel, lubricants, mechanical properties, microhardness

## Abstract

The paper presents the results of an analysis of the process of drawing AISI 321 stainless steel thin-walled seamless tubes on a floating plug. The influence of the geometry of dies and plugs, drawing velocity, and lubricants on the possibility of carrying out the pipe drawing process without a loss of strength of the lubricating film and, consequently, disturbance of the forming process and tube cracking, and also on the temperature in the drawing process, the mechanical properties of the tubes drawn, and the microhardness and roughness of the inner and outer surface of the tubes was investigated. The parameters of the drawing tools used were as follows: angle of drawing dies α = 16° and floating plugs with angles of inclination of the conical part of the plug β = 11.5°, 13°, and 14°. The drawing dies and floating plugs were made of G10 sintered carbide. Drawing speed was varied over the range 1 to 10 m/min. The study used several lubricants. Tubes with dimensions (outer diameter D_0_, wall thickness g_0_ before drawing process) D_0_ = 19 mm, g_0_ = 1.2 mm and D_0_ = 18 mm, g_0_ = 1.2 mm were drawn to produce tubes with dimensions (outer diameter D_k_, wall thickness g_k_ after drawing process) D_k_ = 16 mm, g_k_ = 1.06 mm on a drawbench with the same total elongation, while the diameter and wall thickness were changed. During the process, continuous measurements were made of the drawing force and temperature in the deformation zone and on the tube surface. It was found that the drawing process causes a decrease in the roughness parameters Ra and Rz of the inner surface of the tubes. Moreover, after drawing, an increase of 30–70% was observed in the microhardness of the tube material in relation to the microhardness of the charge material. Based on the test results, it can be concluded that the work of frictional forces is the main direction of optimization of tube drawing on a floating plug process of hard-deforming materials.

## 1. Introduction

Tube drawing using a floating plug is an important and widely-used technology. A characteristic feature of the process is high intensity heat emission in the deformation zone, especially during drawing hard-deforming materials with high yield strength and work hardening [1,2]. Therefore, the potential of the process depends on operations to minimise the amount of heat emission while improving the quality of the lubricant, thus ensuring a high resistance to very high unit pressure and high temperature in the deformation zone [3]. There are three designs of plug that could be of interest to a producer of stainless steel tubing: a ceramic plug, an oscillating plug, and an adjustable plug [4]. The advantage of the floating-plug drawing (FPD) process is the possibility of using a very high velocity of drawing and the ability to achieve a very high productivity [5].

The frictional forces in the FPD process occur at the contact of the outer surface of the tube and the die in the zones of free reduction of diameter, the zone of simultaneous reduction of the diameter and the tube wall, and in the calibration cone of the drawing die. The internal friction surface is the zone of simultaneous reduction of the diameter, the wall of the tube, and the surface of the cylindrical part of the plug in the calibration zone (Figure 1).

The work of frictional forces on the above-mentioned surfaces is completely transformed into heat, resulting in a local increase in the temperature of the drawing die, tube, and floating plug, as well as the lubricating film on the outer and inner surfaces of the tube [6,7]. The intensity of the heat released is a function of the following factors [8,9]:-kind of tube material drawn, its yield point, and the intensity of work hardening,-friction conditions and tribological properties of the lubricant used,-drawing speed,-geometric parameters of the deformation zone (the angle of the die cone, the angle of the conical zone of the floating plug, the material strain in the diameter reduction zone and in the zone of simultaneous diameter reduction and wall thinning, and the width of the calibration strip.-A feature that distinguishes the process of drawing tubes with a floating plug is the high speed of drawing tubes made of materials with a low yield point, low work hardening intensity, and the need to reduce the drawing speed of hard-to-deform materials.

Analysis of the issues shows that previously proposed solutions related to the influence of various factors on the stability of the plug in the deformation zone [10,11], the impact of dependency and the geometrical dimensions of dies and plugs on the intensification of deformation and process velocity [5,6,7,10,12,13], tool materials [6], minimising stress during drawing [14,15], and the influence of friction and lubricants applied by various methods in carrying out the process [16,17,18,19]. After introducing the technology of tube drawing with a floating plug into production, it was often believed that tube drawing was only possible under the condition that D_T_ > D_1_ (where D_T_ is the plug head diameter and D_1_ is the diameter of the die) [20]. However, it was found [7,21] that thick-wall tubes can be drawn, but only under the condition that D_T_ < D_1_.

Floating plug drawing has been the research topic of many authors in terms of the improvement of the shape and dimensional accuracy of tubes, ensuring adequate lubrication, the improvement of the surface finish, and the minimisation of drawing forces [22]. Danckert and Endelt [23] proposed a new plug with a circular profiled plug instead of a conventional cylindrical plug, which forms a cylindrical bearing channel between the die and the plug. The results of numerical FE-based analysis showed that the drawing force can be reduced and that the drawing force is nearly independent of the length of the die and of small variations in the angle of the die. Damodaran et al. [24] recommended a short bearing channel in order to increase the stability of the drawing process. Yoshida and Furuya [25] determined an optimal shape for the plug using the finite element method (FEM). They investigated the fabrication of fine tubes used in catheters and stents by means of plug drawing and mandrel drawing. Experiments using a ball-type plug, different from the conventional conical plug, allowed the frequency of breaks during drawing to be reduced. Wang et al. [26] proposed a new mathematical model to predict the behaviour of tube drawing with a floating plug in hollow copper tubes. This model incorporates the plug semi-angle, die semi-angle, wall thickness reduction ratio, and friction coefficient. The results confirmed that the model developed is more accurate than the conventional slab model, especially at reduction ratios of up to 40%. Şandru and Camenschi [27] developed a theory of high speed tube drawing with a floating plug. Schrek et al. [28] focused on an analysis of factors negatively affecting the FPD of tubes made from austenitic stainless steel. The results obtained showed very good agreement between the simulations and the experimental results.

Partial or comprehensive research on the process of drawing tubes on a floating plug was carried out for relatively easily deformable materials (aluminium [23], copper [29,30], brass [15,31,32,33], cobalt–chromium alloys [34], carbon and alloy steel [23,35]) and with the use of tools and lubricants appropriate for these materials and drawing speeds. These studies do not answer a number of questions about the same process using hard-to-deform materials.

In order to reveal the effect of the area of the friction contact surface on the work of external frictional forces and the surface temperature, it was decided to change the difference in the cone angles of the die α and the angle of inclination of the conical part of the plug β within the range (α − β) = 2° − 4.5°, determined on the basis of previous research. Literature analysis and information collected from lubricant manufacturers made it possible to select the appropriate lubricants. Strain gauge measurement of the drawing force, continuous multi-point measurement of the surface temperature of the die, and contact measurement of the temperature of the tube surface behind the drawing die were the tasks selected to evaluate the influence of the parameters of the drawing process on the value of the work of the friction force. An evaluation of the influence of the parameters of the FPD process on tubes was made on the basis of measurements of surface roughness, microhardness, and the mechanical properties of the tubes.

## 2. Experimental

Drawing tests were carried out on AISI 321 steel tubes with the following outer diameter D_0_ and wall thickness g_0_ (D_0_ = 19 mm, g_0_ = 1.2 mm, D_0_ = 18 mm, g_0_ = 1.2 mm). The outer surface of the tubes was prepared by grinding after etching (Figure 2a,b). These treatments were carried out to create lubricant pockets on the outer surface of the tube, allowing more lubricant to be delivered to the deformation zone. The purpose of the application of this treatment was to improve the lubrication conditions, which resulted in the reduction of the amount of heat released during the drawing process, as well as the reduction of the irregularity of the deformation and a reduction of the residual stresses.

Tubes were drawn on a floating plug using a drawing machine with a maximum drawing force of 70 kN. The final outer diameter of tubes was D_k_ = 16 mm, and the wall thickness was g_k_ = 1.06 mm. The elongation ratios in diameter were λ_D_ = 1.19 and λ_D_ = 1.12, and the elongation ratios in wall thickness were λ_g_ = 1.13 and λ_g_ = 1.2. Maintaining the same total elongation values at different elongation ratios resulted in the reduction of the diameter and the tube wall thickness, giving the possibility of differentiating the total friction surface and revealing the stress in the tube material due to the reduction in diameter. As a drawing tool, a tungsten carbide G10 die was used with a half angle of α = 16° and a diameter of D = 16 mm.

Drawing dies were fabricated with special holes on the ring to allow the temperature to be measured (Figure 2c,d) at four points in the deformation zone using thermocouples. During the trials, the temperature of the outer surface of the tube was also continuously measured at the outlet of the die using a Flir TG54 pyrometer (Teledyne FLIR Company). Floating plugs were made of G10 tungsten carbide with a half angle of the angle of inclination of the conical part of the plug β = 11.5°, 13°, and 14°, which enabled various angles between the die and the plug to be obtained in the range (α − β) = 2° − 4.5° (Figure 2e–h).

During the tests, the drawing velocity was varied in the range 1–10 m/min. WISURA DSO 7010, Fuchs Oil Corp., further in the text marked as (W), Prolong EP 2, Prolong Super Lubricants (P), Ferokol EPS-220, Naftochem, Corp. (F), Masterdraw 7194AC, ETNA Products, Inc. (M), and Tubol 1962JR, Metalube Limited (T), lubricants were used in the drawing experiments. The lubricants selected for testing are recommended for drawing tubes by world-renowned manufacturers. To measure the force and the temperature, a measuring circuit consisting of a force sensor, a Spider 8 Hottinger Baldwin Messtechnik measuring amplifier, K-type thermocouples, and a personal computer Dell Latitude 5520 were used. The outer diameter of the pipes was measured using an electronic calliper. The wall thickness was measured using an electronic micrometer. In addition, for comparison and verification, the outer diameter of the tubes and the wall thickness were measured using the Atos Core 200 optical 3D scanner from GOM company. The surface topography and roughness of the inner and outer surfaces of the tubes were measured using LEXT OLS4100 confocal laser microscope with 405 nm UV laser light (Leica Microsystems). Surface roughness results are presented by the two most commonly used parameters, i.e., Ra—arithmetic average of the absolute values of the profile heights over the evaluation length and Rz—the average value of the absolute values of the heights of five highest-profile peaks and the depths of five deepest valleys within the evaluation length. The ground surface of the batch material and the surfaces of the tubes after drawing were analysed. Vickers microhardness measurements were made using a Shimadzu HMV-2T tester. The investigations of mechanical properties were carried out using a Zwick Roell Z005 uniaxial tensile test machine. The microstructure studies were carried out on the Olympus GX51 light microscope. The test specimens were cold mounted in a Struers FixiForm container using Struers EpoFix epoxy resin. The samples prepared in this way were then ground on sandpaper with a grit of 240–2000. For polishing, diamond suspensions were used: DP-Suspension P 9 μm, 3 μm, 1 μm. The specimens were made on a RotoPol-11 grinder–polisher with a RotoForce-1 Struers head. Finishing polishing was carried out with the OP-S polishing slurry from Struers. To reveal the AISI 321 stainless steel microstructure, the specimens were electrolytically etched in 10% nitric acid at a voltage of 1.5 V for 20 s at room temperature.

## 3. Results and Discussion

### 3.1. Drawing Force and Temperature

The results of the drawing force and the temperature at various angles β between the die and the floating plug (a drawing velocity v_c_ in the range of 1–10 m/min with (W) lubricant) are presented in Table 1 and Table 2. An increase in the drawing velocity caused an increase in the value of the drawing force, reaching a value of F_max_ = 30.15 kN for a drawing velocity of v_c_ = 6 m/min (Table 1).

A large friction surface between the plug and the inner surface of the tube was associated with differences in the angles of the die α and the angle of inclination of the conical part of the plug β (α − β) equal to 4.5°. This causes an increase in unit pressure and a deterioration in friction conditions. In addition, for such a difference in the angles, too big a gap is formed between the barrel surface of the plug and the inner surface of the tube, preventing the development of a hydrodynamic effect at low drawing velocity.

Graphs of the course of the temperature change and drawing strength allow one to assess the stability of the drawing process as a function of time, and they also contain information about disturbances, which prove to be linked to the interruption of the lubricant film. Analysing the maximum and average drawing force and the tube surface after drawing using the lubricant W showed that the “bamboo” effect occurred with drawing conditions α = 16°, β = 11.5°, and v_c_ = 1 m/min. A partial “bamboo” effect occurred with velocity v_c_ = 2 m/min (Figure 3). On the other hand, there was no such effect for velocity v_c_ = 3 or 4 m/min (Figure 4) and 6 m/min. This may indicate a major impact of drawing velocity, die geometry, and the floating plug and temperature emissions on the occurrence of instabilities during the process.

Variation in temperature and drawing force for the die angle α = 16° and plug angles β = 13° and 14° are similar to the curves shown in Figure 4. When drawing using dies with an angle α = 16° and plug angle β = 13°, first, a clear decrease and then an increase in drawing force occur with an increase in drawing velocity. The strongest force F_max_ = 34.1 kN was for the velocity v_c_ = 1 m/min, while the weakest force F_max_ = 29.1 kN was for a velocity v = 2 m/min (Table 1). For the velocity v_c_ = 4 m/min, there is an increase in drawing force to the value of F_max_ = 31.3 kN. For angles α = 16° and β = 13° and velocity v_c_ = 2 m/min, the average drawing force is F_av_ = 25.2 kN. For drawing with a die angle α = 16° and plug angle β = 14°, an increase in drawing velocity causes a decrease in the maximum drawing force, which takes the value F_max_ = 28.6 kN for v_c_ = 1 m/min and F_max_ = 25.3 kN for velocity v_c_ = 4 m/min (Table 1).

Analysis of the temperature occurring in the die and on the tube surface indicates that, for α = 16° and β = 11.5°, 13°, and 14°, there is an increase in those temperatures with increasing drawing velocity. The lowest maximum temperatures, T_max_ = 69.5 °C in the die and T_max_ = 77.7 °C on the tube surface, were measured for the angle α = 16° and β = 14° with a drawing velocity v_c_ = 1 m/min (Table 2), while the highest maximum temperatures, T_max_ = 84.4 °C in the die and T_max_ = 133.2 °C on the tube surface, were recorded for α = 16° and β = 11.5° at a drawing velocity v_c_ = 6 m/min. This demonstrates that the parameter having the largest impact on the temperature emitted during the process is that of drawing velocity. A small temperature difference between the die and the tube surface (15 °C for α = 16°, β = 13°, and v_c_ = 1 m/min) proved sufficient for the die and lubricant to release heat. With an increase in drawing velocity, a distinct difference was observed between the temperature in the die and on the tube surface (50 °C for α = 16°, β = 11.5°, and v_c_ = 6 m/min). This indicates that it is impossible for both the die and the lubricant to give up heat due to the lack of die cooling and the fact that the heat capacity of the lubricant is too small.

Temperature change tends to increase during drawing. This increase is greater at lower drawing velocity. With an increase in drawing velocity, the curves are flatter. But, they failed to achieve a constant temperature during the drawing of tubes with lengths of 2000 mm, thus providing the conditions for a stationary process.

When the “bamboo” effect occurred, which presented a periodic build-up on the tube surface (Figure 5a), an oscillation of temperature was observed (Figure 3). The temperature curves are composed of characteristic “teeth”, corresponding to the oscillations of the drawing force, starting and ending with the “bamboo” effect.

During tube drawing on dies with angles α = 16° and β = 11°30′, 13°, and 14° over the complete range of drawing velocity using lubricants (P) and (F), unsatisfactory results were obtained. Tube breakages were observed after 50 s from the start of the drawing process, i.e., after the tube was drawn for about 1.5 m (Figure 5b). The occurrence of tube breakages is a problem during drawing, with an approximate 20% rate in production [24].

A characteristic feature of the loss of lubricant properties was a clear increase in the temperature in the die and a corresponding increase in force (Figure 6a). Thus, a clear deterioration in physical properties and the viscosity of the lubricant under the influence of an increase in temperature can be observed in the deformation zone, revealing a rupture of the lubricant film and an increase in the drawing force.

Completely unsatisfactory results were obtained when drawing tubes with a dimension D_0_ = 19 mm, g_0_ = 1.2 mm using a die with dimensions α = 16° and β = 11.5°, 13°, and 14° while using the lubricants (M) and (T) (Figure 6b).

When analysing the influence of the strain values applied, it should be noted that positive test results were obtained for the drawing of tubes with dimensions D_0_ =19 mm, g_0_ = 1.2 mm into tubes with dimensions D_k_ = 16 mm, g_k_ = 1.06 mm, while unsatisfactory results were obtained for the drawing of tubes from dimensions D_0_ = 18 mm, g_0_ = 1.2 mm into dimensions D_k_ = 16 mm, g_k_ = 1.06 mm, with the same total strain amounting to λ = 1.34. The difference was in the partial deformations of the wall thickness and diameter of the tube. Satisfactory results were obtained for the drawing of tubes from dimensions D_0_ = 19 mm, g_0_ = 1.2 mm into dimensions D_k_ = 16 mm, g_k_ = 1.06 mm, because it was possible to obtain conditions allowing the production of natural back-tension, thus lowering the pressure of the metallic workpiece on the die and the plug with no increase in the drawing force.

The intended use of the lubricants used is strictly defined, not only in terms of strength properties, viscosity as a function of temperature, and the wettability of the surfaces on the tube, drawing die, and floating plug, but also due to the use of a specific material with specific properties. The (W) lubricant can be successfully used for drawing tubes made of AISI 321 steel on a floating plug. On the other hand, (T) and (M) lubricants did not fulfil their task completely when drawing this type of austenitic steel, despite the fact that they are successfully used for drawing copper and brass.

The parameter determining the feasibility of the FPD process is the strength of the lubricating film on the external and internal friction surfaces. The strength of the lubricant film depends on the type of lubricant and the concentration of surface-active agents in the lubricant, and, most of all, on the temperature in the friction zones. If the critical temperature is exceeded, the lubricating film is ruptured and the tube is almost immediately ruptured. Thus, in the case of drawing tubes made of both soft and hard-to-deform materials, the necessary condition for the drawing process is to ensure that temperatures in the deformation zone are lower than the critical temperature for the lubricant used. Optimisation of the geometry of the tools in order to ensure that the minimum work of frictional forces is fully converted into heat will minimise the drawing stress, permitting the FPD process to be carried out.

### 3.2. Microstructure

Table 3 shows examples of the topography of the outer and inner surfaces, as well the microstructure, on the cross section of tubes before and after drawing with β = 11.5°, 13°, 14°, α = 16°, v_c_ = 4 m/min, and with the use of (W) lubricant.

Microstructural observations of the charging tubes revealed large grains in the central part of the cross-section, with refined grains on the inner and outer surfaces of the tubes. This indicates a poor selection of heat treatment parameters (supersaturation) in the process of preparing the material for drawing. This is also confirmed by the results of the average microhardness measurements on the cross-section of the charge tubes. Microhardness in the middle part of the wall thickness was 160–173 HV, while the microhardness in the vicinity of the inner and outer surfaces of tube was 180–190 HV.

Grinding the outer surface of the charge tubes causes the formation of lubricant pockets that allow more lubricant to be supplied to the deformation zone, improving the frictional conditions of the process (reducing the temperature and the amount of heat generated during drawing). Microscopic observation showed that the scratches forming transverse to the tube axis after grinding remain visible on the surface of the product regardless of the drawing process conditions; however, they are clearly flattened. Longitudinal scratches visible on the inner and outer surfaces of the finished product indicate a lack of the appropriate amount of lubricant or the fact that the fine particles remaining after the grinding process caused a roughening of the tube surface. Varying the geometry of the tools used for drawing and changing the speeds of the process do not significantly change the microstructure of the tubes.

### 3.3. Surface Roughness

#### 3.3.1. Experimental Results

The results of the mean roughness Ra and Rz measured on the inner and outer surfaces of the tubes, depending on the angle of the floating plug and the drawing speed, are presented in Table 4 and Figure 7 and Figure 8. Surface roughness parameters were measured on the tubular surfaces in the axial direction.

The roughness measurements of the tubes after drawing confirmed the significant influence of the process parameters on the surface roughness of the inner and outer surfaces of the tubes. After the drawing process, there was a significant improvement in the smoothness of the (especially inner) surface of the tubes in relation to the smoothness of the surface of the charging tubes. In the case of the charging tubes, the mean roughness of the internal surface was Ra = 1.25 μm, and, for the external surface, was Ra = 0.7 μm. After drawing, the mean roughness of the tube surface was between Ra = 0.43 μm and Ra = 0.93 μm for the outer surface of the tubes and Ra = 0.09–0.35 μm for the inner surface of the tubes. The mean roughness and 10-point peak–valley surface roughness of the inner surface of the tubes was significantly reduced (Figure 7). The drawing process led to a reduction in the surface roughness parameters Ra and Rz measured on the outer surface of the tubes, but only for the angles of floating plug β = 13° and β = 14° (Figure 8).

With the use of the die angle α = 16° and the angle of floating plug β = 11.5°, 13°, and 14°, along with the increase in the drawing speed, the roughness of the inner surface of the tube increases, which demonstrates the improvement of the lubrication conditions and the increase in the thickness of the lubricating film. FPD of tubes significantly improves the roughness of the inner surface of the tubes in relation to drawing the tubes without a floating plug, in which the circumferential compressive stresses promote an increase in the roughness of the inner surface. The contact of the inner surface of the tube with the floating plug leads to a radical improvement in roughness. It can be concluded that the increase in speed and angle of the die increases the effect of hydrodynamic drawing, which results in a reduction of the friction coefficient that mainly occurs on the inner surface of the tube [12,13].

#### 3.3.2. Analysis of Variance

First, analysis of variance was used to model the relation between the angle of floating plug β, the drawing speed v_c_, the measurement side (inner and outer surfaces of tubes), and the value of the roughness parameter Ra. THe measurement side is considered to be a categoric variable.

The model F-value of 66.04 implies the model is significant (Table 5). The model terms are significant when *p*-values are less than 0.0500. In this case, B and C are significant terms. Values greater than 0.1000 indicate the model terms are not significant. A signal-to-noise ratio (an adequacy precision ratio) of 21.288 indicates an adequate signal for the model (Table 6). If the adequacy precision is greater than 4, the ANOVA model can be used to navigate the design space. The predicted R^2^ of 0.8178 is in statistically reasonable agreement with the adjusted R^2^ of 0.8610.

The relation between the input parameters and the roughness parameter Ra (in terms of the coded factors) is as follows:(1)Ra =0.4400+0.1161A+0.1586B

The analysis of variance was also used to model the relation between the angle of floating plug β, the drawing speed v_c_, the measurement side (inner and outer surfaces of tubes), and the value of the roughness parameter Rz. The measurement side is considered to be a categoric variable.

The model F-value of 69.94 implies the model is significant (Table 7). The model terms are significant when *p*-values are less than 0.0500. In this case, C, AB, B^2^ are significant terms. Values greater than 0.1000 indicate the model terms are not significant. A signal-to-noise ratio (an adequacy precision ratio) of 22.6581 indicates an adequate signal for the model (Table 8). If the adequacy precision is greater than 4, the ANOVA model can be used to navigate the design space. The predicted R^2^ of 0.9274 is in statistically reasonable agreement with the adjusted R^2^ of 0.9426.

The relation between the input parameters and the roughness parameter Rz (in terms of the coded factors) is as follows:(2)$$\mathrm{Rz}\;=4.18-0.5062A+0.1367B+1.48C-1.04\mathrm{AB}-1.01B^{2}$$

### 3.4. Microhardness

#### 3.4.1. Experimental Results

The results of microhardness measurements of the tubes show a significant increase of 30–70% in the microhardness value on the cross-section of the tubes after drawing, in relation to the microhardness of the charge material. A characteristic feature is the differentiation in the amount of microhardness across the thickness of the wall. In the middle zone of the wall, this increase in value is less than for the values in the outermost zones. The differentiation of angles of the dies and floating plugs, as well as the drawing speed, do not significantly affect the microhardness value (Figure 9).

The average microhardness of the tubes after drawing is in the range of 240–310 HV. Higher values of microhardness 250–310 HV occur for the die angle α = 16°. With the increase in the difference (α − β), microhardness values increase—the largest occur for α = 16°, β = 11.5° (α − β = 4.5°) (Figure 10). This is due to the greater share of bending forces at tube entry and at the exit of the tube from the die.

#### 3.4.2. Analysis of Variance

The model F-value of 12.93 implies the model is significant (Table 9). The model terms are significant when *p*-values are less than 0.0500. In this case B, A^2^, C^2^ are significant terms. Values greater than 0.1000 indicate the model terms are not significant. A signal-to-noise ratio (an adequacy precision ratio) of 12.8640 indicates an adequate signal for the model (Table 10). If the adequacy precision is greater than 4, the ANOVA model can be used to navigate the design space. The predicted R^2^ of 0.4470 is in statistically reasonable agreement with the adjusted R^2^ of 0.5248.

The relation between input parameters and microhardness (in terms of the coded factors) is as follows:(3)$$\mathrm{HV}=254.11-0.3038A+6.38B-0.8455C+7.02A^{2}+23.89C^{2}$$

### 3.5. Mechanical Properties

The ultimate tensile stress of the tubes after drawing is almost doubled as compared to the charge material (Figure 11). On the other hand, plastic properties decrease more than three times (Figure 12), which proves a significant work hardening of the material has taken place during the drawing process.

There is no clear correlation between the parameters of the drawing process and the results obtained for the mechanical properties. When analysing the influence of the floating-plug drawing process on the mechanical properties of the material, it should be noted that, for the total deformation of λ = 1.34, the strength properties increase from the level of R_m_ ≈ 560 MPa for the charge material to R_m_ ≈ 900−960 MPa for the tubes after drawing, with a simultaneous decrease in elongation from A_5_ = 62% to A_5_ = 12.5–20%. This demonstrates a huge increase in the strengthening of cold-processed material with a decrease in plastic properties.

#### Analysis of Variance

In this section, two models of ANOVA were considered. First, analysis of variance was used to model the relation between the angle of the floating plug β, the drawing speed v_c_, and the value of uniaxial tensile strength R_m_. In the second model, the elongation A_5_ was used as the output. In both cases, the model F-values imply that the models are not significant relative to the noise. Therefore, the influence of the input parameters on the two analysed mechanical parameters can be described by the average value of these parameters.

## 4. Conclusions

The following conclusions can be drawn on the basis of the analysis of the results obtained:-The application of an appropriate deformation in the reduction zone of the drawing die results in the creation of back tension, which reduces the pressure of charge metal on the die and plug.-The reduction of unit pressure, as a consequence, reduces the work of forces of external friction.-The use of higher values of the angles of the die and floating plug, while maintaining the difference between the angles (α − β) = 4.5°, reduces the friction surface and the work of friction forces.-An increase in the drawing speed, while maintaining an appropriate gap between the inner surface of the tube and the cylindrical part of the floating plug, significantly improves the friction conditions due to the increasing effect of the hydrodynamic lubrication phenomenon.-An increase in the roughness of the inner surface, with an increase in the drawing speed for high values of angles of the die and plug, demonstrates the hydrodynamic effect during the drawing process.-It is recommended that the materials used for tools should guarantee a small value of coefficient of friction.-It is recommended that lubricants be used that have tribological properties that create conditions ensuring adequate strength of the lubricating film at a temperature of 90–150 °C.-The drawing process causes a decrease in the roughness parameters Ra and Rz of the inner surface of the tubes.-The surface roughness of the outer surface of the tubes does not change much in relation to the surface roughness of the charging tubes. The lowest values of the Ra and Rz surface roughness parameters measured on the outer surface of the tubes occur for the die angle α = 16° and the floating plug angle β = 13°.-For the die angle α = 16° and plug angles β = 11.5°, 13°, and 14°, the values of the roughness parameters increase with an increase in drawing speed, although this increase is small. This demonstrates the increase in the thickness of the lubricating film and, thus, the improvement of the lubrication conditions.-After drawing, an increase of 30–70% was observed in the microhardness of the tube material in relation to the microhardness of the charge material. In the middle zone of the wall thickness, the increase in microhardness is less than in the outer zones. The different angles of the dies and plugs, as well as the drawing speed, do not significantly affect the microhardness value.-The value of ultimate tensile stress doubled compared to the property in the charge material. However, the plastic properties decreased by more than three times due intensive strengthening of the material. At the same time, no correlation was observed between the drawing process parameters and material properties.-Limitations of the floating-plug pipe drawing method result from plug geometry and improperly selected lubricants.-Future works will focus on the possibility of increasing the drawing speed and on the selection of both lubricants and lubrication methods, which determine the possibility of intensifying and increasing the efficiency of the drawing process.

## Figures and Tables

**Figure 1 materials-16-05684-f001:**
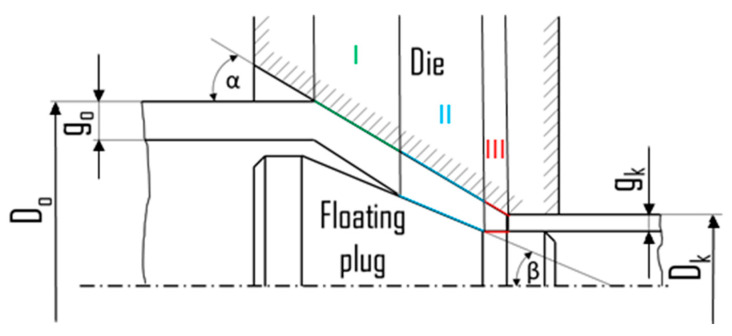
Schematic diagram of the FPD process with marking free reduction of diameter zone (I), drawing zone on the conical (II) and cylindrical (III) plug parts, outer diameter and wall thickness of the tube before (D_0_, g_0_) and after (D_k_, g_k_) the drawing process, as well as die (α) and plug (β) angles.

**Figure 2 materials-16-05684-f002:**
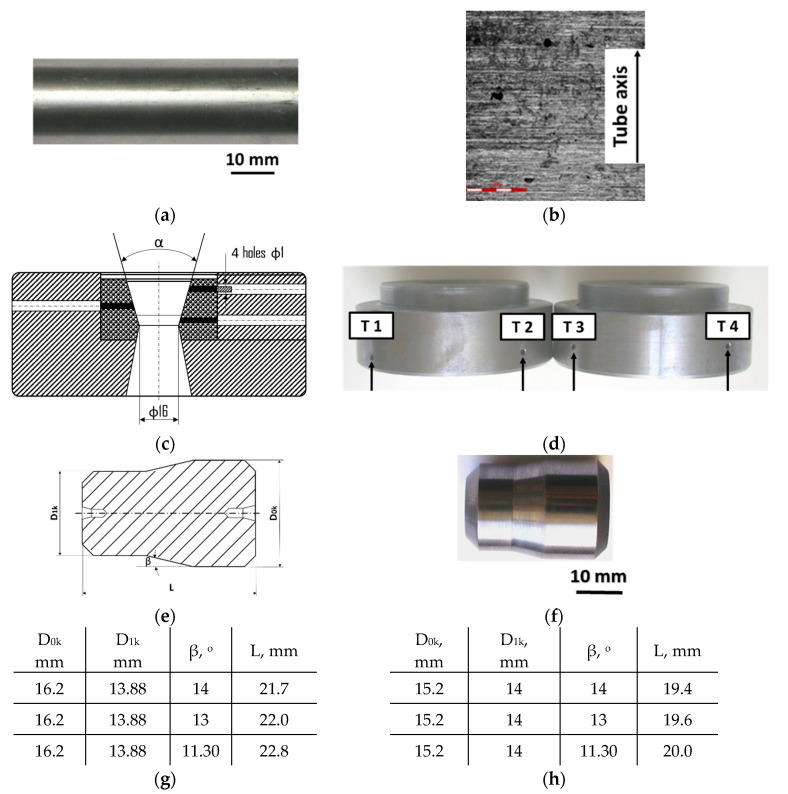
AISI 321 stainless steel tube with ground surface: (**a**) macro and (**b**) micro observation, (**c**) drawing die with holes to allow temperature measurement in the deformation zone, (**d**) view of dies with holes drilled in them, (**e**) schematic drawing of a floating plug, (**f**) sample of floating plug, floating plugs dimension according to the scheme of tube drawing (**g**) D_0_ = 19 mm, g_0_ = 1.2 mm and D_k_ = 16 mm, g_k_ = 1.06 mm, and (**h**) D_0_ = 18 mm, g_0_ = 1.2 mm and D_k_ = 18 mm, g_k_ = 1.06 mm.

**Figure 3 materials-16-05684-f003:**
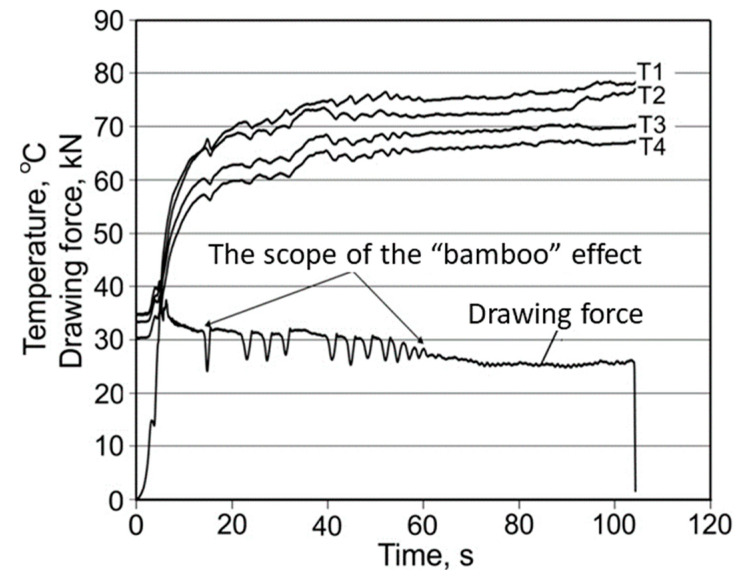
Evolution of both the temperature of die and drawing force F_c_ for α = 16°, β = 11.5°, lubricant: W, v_c_ = 2 m/min.

**Figure 4 materials-16-05684-f004:**
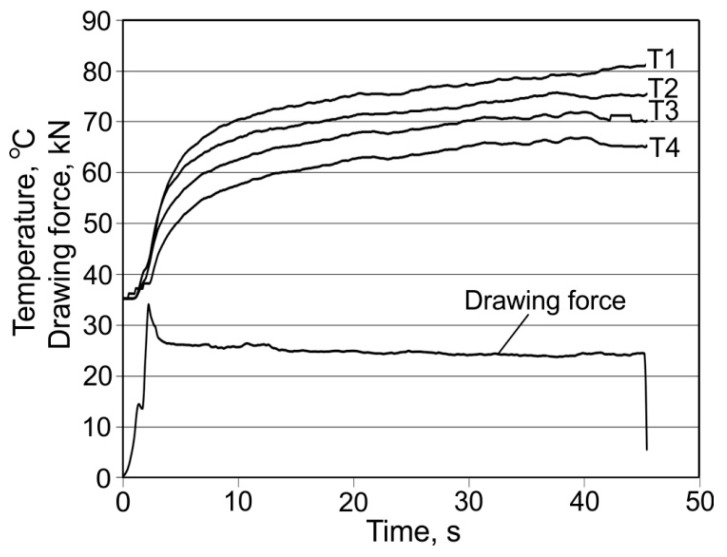
Evolution of both the temperature of the die and drawing force F_c_ for α = 16°, β = 11.5°, lubricant (W), v_c_ = 4 m/min.

**Figure 5 materials-16-05684-f005:**
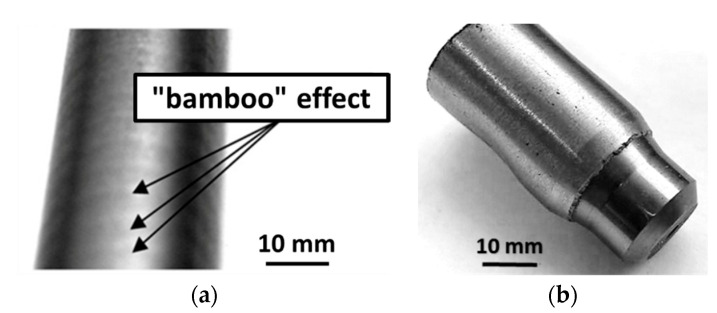
(**a**) “Bamboo” effect formed on the tube surface as a result of the instability of the process conditions, and (**b**) view of the broken tube with a floating plug inside.

**Figure 6 materials-16-05684-f006:**
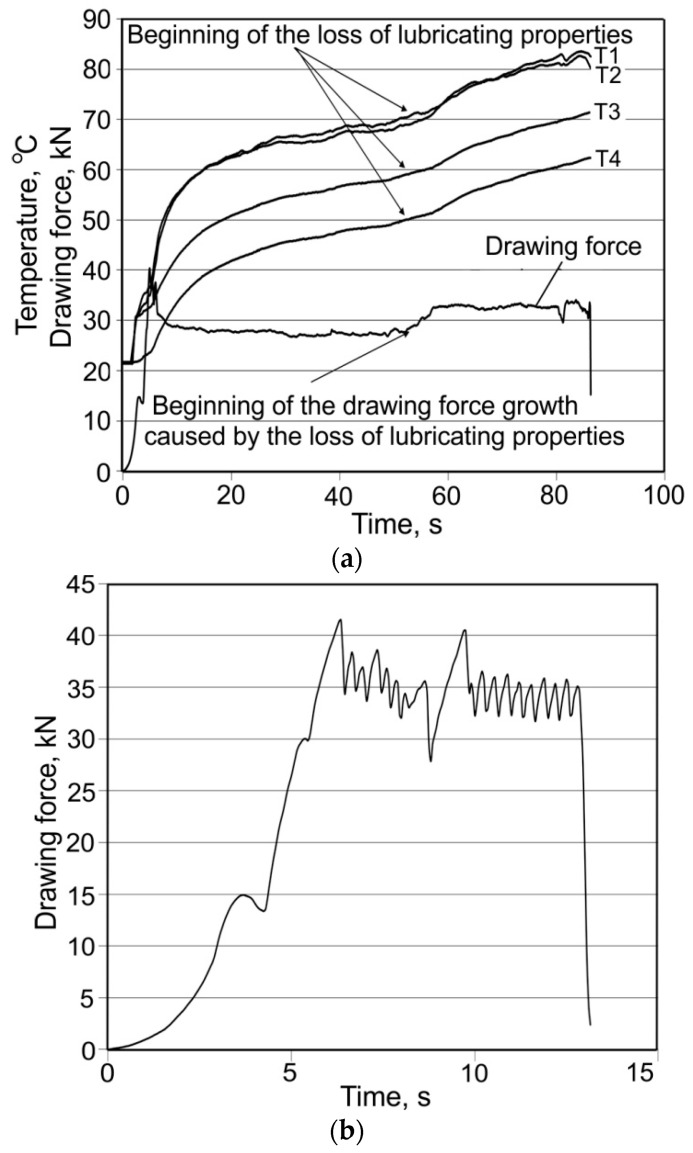
Variation of temperature and of drawing force during drawing at α = 16°, β = 14°, v_c_ = 2 m/min, and with the use of (P) lubricant (**a**) and (T) lubricant (**b**).

**Figure 7 materials-16-05684-f007:**
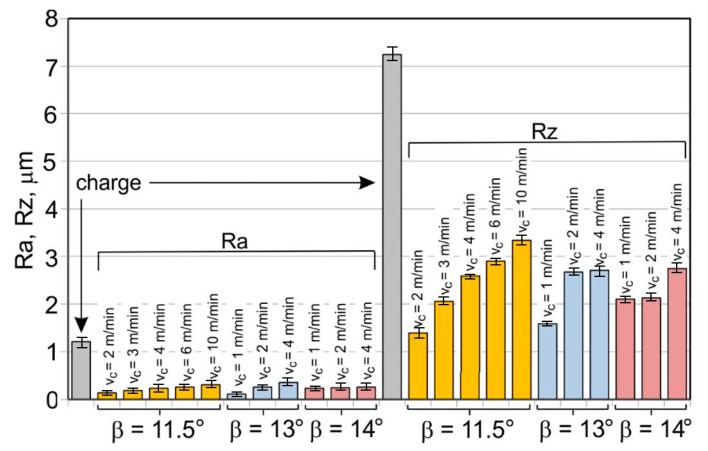
Micro-geometry of the inner surface of the tubes (D = 19 mm, α = 16°, (W) lubricant).

**Figure 8 materials-16-05684-f008:**
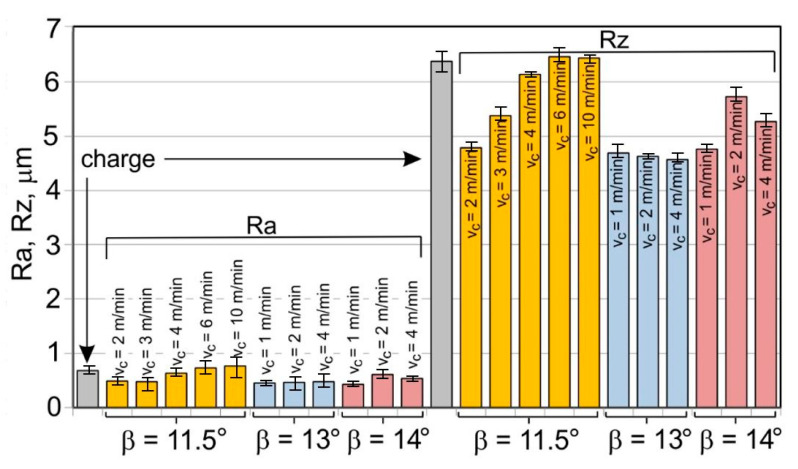
Micro-geometry of the outer surface of the tubes (D = 19 mm, α = 16°, (W) lubricant).

**Figure 9 materials-16-05684-f009:**
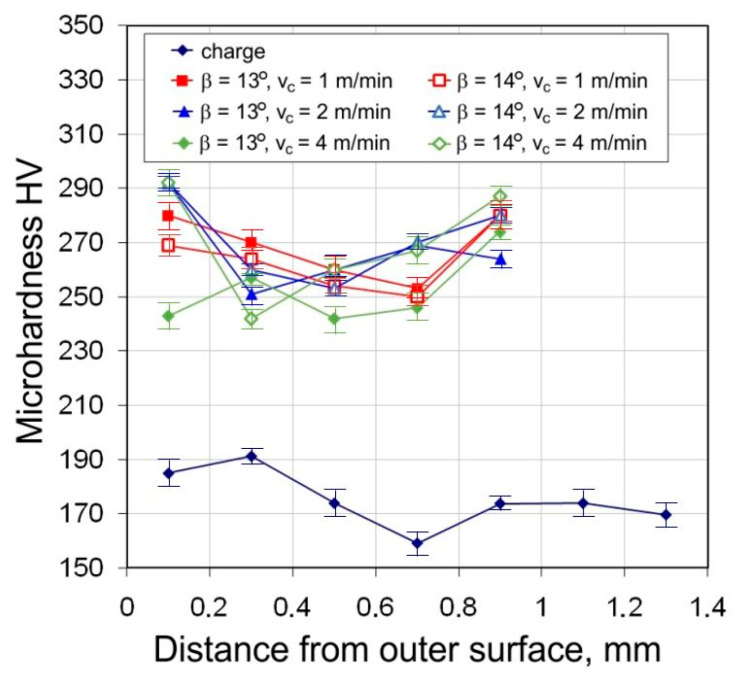
Micro-hardness across the tube cross section (D = 19 mm, α = 16°, (W) lubricant).

**Figure 10 materials-16-05684-f010:**
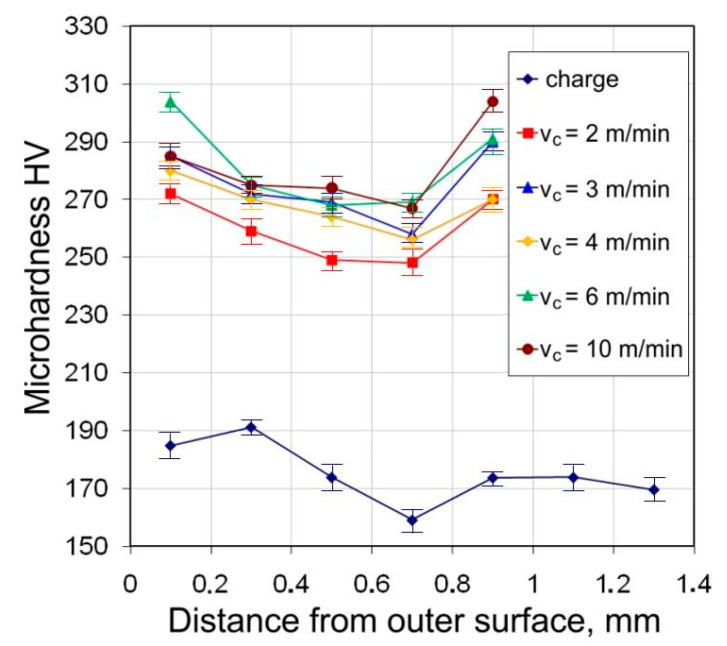
Effect of drawing speed v_c_ on the microhardness across the tube cross section (D = 19 mm, α = 16°, β = 11.5°, (W) lubricant).

**Figure 11 materials-16-05684-f011:**
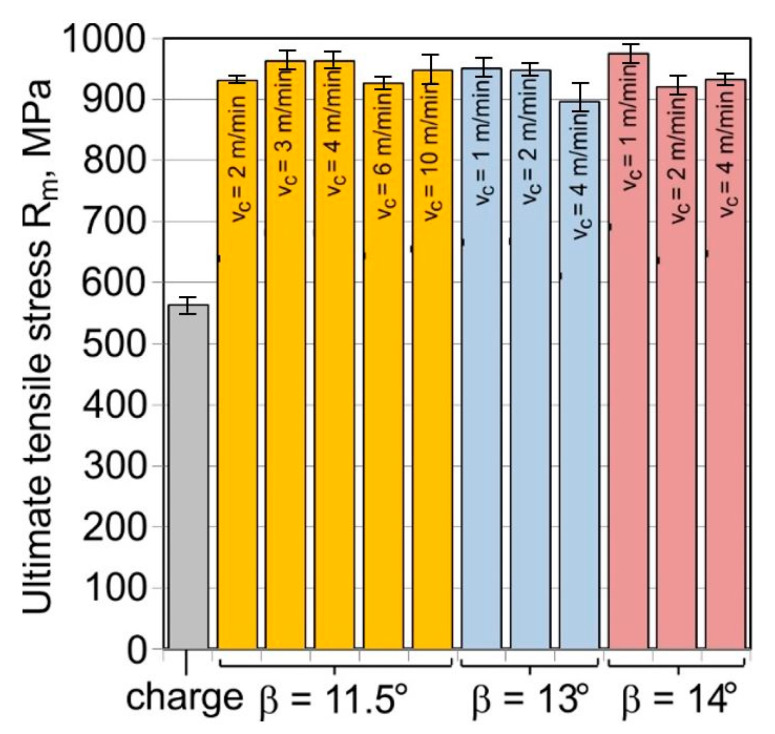
Ultimate tensile strength R_m_ of the tube material (D = 19 mm, α = 16°, (W) lubricant).

**Figure 12 materials-16-05684-f012:**
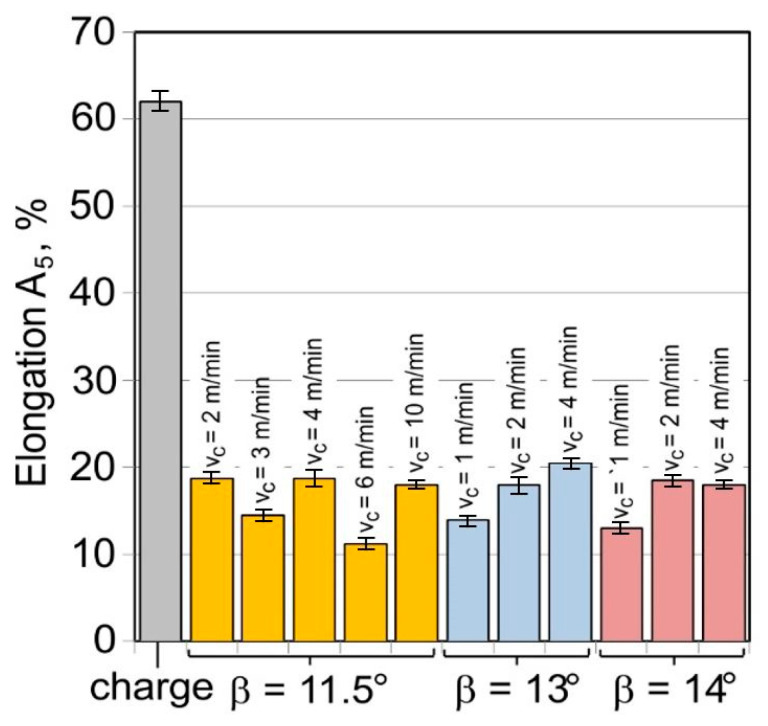
Elongation A_5_ of the tube material (D = 19 mm, α = 16°, (W) lubricant).

**Table 1 materials-16-05684-t001:** Drawing forces and maximum temperature of tube and drawing die—D = 19 mm, α = 16°, (W) lubricant.

β, °	v_c_, m/min	F_max_, kN	F_śr_, kN	T_max_ (Drawing Die), °C	T_max_ (Tube), °C	Comment
11.5	1	41.5	34.9	-	-	The phenomenon of periodic pulsation of force (the so-called “bamboo” effect). Tube breakage.
11.5	2	27.2	25.7	78.3	112.3	Partial phenomenon of periodic pulsation of force (the so-called “bamboo” effect).
11.5	3	27.5	25.6	77.8	104.8	
11.5	4	27.7	24.8	81.4	109.4	
11.5	6	30.2	27.6	84.4	133.2	
11.5	10	28.6	26.7	78.3	111.1	
13	1	34.1	29.1	70.2	85.1	
13	2	29.1	25.3	74.4	101	
13	4	31.3	27.3	83.9	123.1	
14	1	28.6	26.4	69.5	77.7	
14	2	27.4	24.6	75.9	96.9	
14	4	25.4	24.9	77.8	111.2	

**Table 2 materials-16-05684-t002:** Temperature of die in points T1–T4—D = 19 mm, α = 16°, (W) lubricant.

β, °	v_c_, m/min	T_1_, °C	T_2_, °C	T_3_, °C	T_4_, °C
11.5	1	-	-	-	-
11.5	2	78.2	77.1	70.4	67.4
11.5	3	77.7	73.8	71.5	68.7
11.5	4	81.4	75.5	71.8	65.3
11.5	6	84.4	78.4	71.9	68.7
11.5	10	76.5	74.5	58.8	54.9
13	1	70.2	70.2	66.7	64.9
13	2	74.4	74.7	71.5	65.6
13	4	83.9	78.1	74.8	69.2
14	1	69.5	68.5	67.1	64.3
14	2	75.9	74.5	67.6	65.2
14	4	77.8	76,1	69.2	66.4

**Table 3 materials-16-05684-t003:** Topography of outer (a, d, g, j) and inner (b, e, h, k) surfaces with roughness measurement lines, as well microstructure (c, f, i, l) of tubes before and after drawing with β = 11.5°, 13°, 14°, α = 16°, v_c_ = 4 m/min, and with the use of (W) lubricant.

	Topography of Outer Surface of Tubes	Topography of Inner Surface of Tubes	Microstructure
Charge	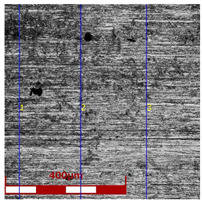	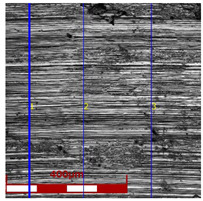	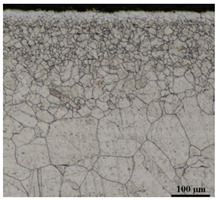
	(**a**)	(**b**)	(**c**)
β = 11.5° α = 16° v_c_ = 4 m/min	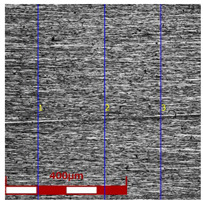	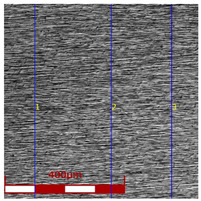	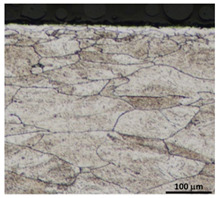
	(**d**)	(**e**)	(**f**)
β = 13° α = 16° v_c_ = 4 m/min	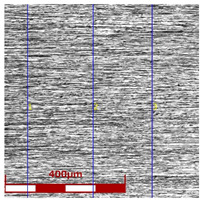	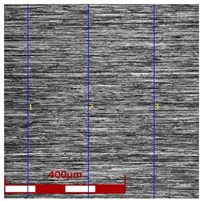	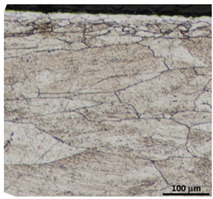
	(**g**)	(**h**)	(**i**)
β = 14° α = 16° v_c_ = 4 m/min	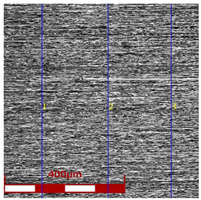	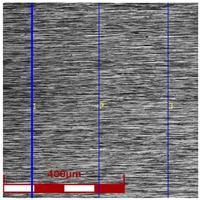	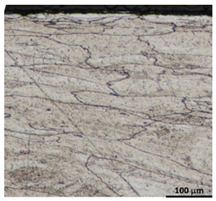
	(**j**)	(**k**)	(**l**)

**Table 4 materials-16-05684-t004:** Roughness parameters Ra and Rz of inner and outer surfaces of tubes—D = 19 mm, α = 16°, (W) lubricant.

Angle of Floating Plug β, °	Drawing Speed v_c_, m/min	Outer Surface		Inner Surface	
		Ra, μm	Rz, μm	Ra, μm	Rz, μm
11.5	1	-	-	-	-
	2	0.491	4.794	0.136	1.391
	3	0.475	5.382	0.185	2.067
	4	0.639	6.146	0.233	2.762
	6	0.734	6.471	0.267	2.904
	10	0.762	6.455	0.302	3.348
13	1	0.458	4.699	0.116	1.583
	2	0.461	4.631	0.264	2.677
	4	0.475	4.572	0.353	2.711
14	1	0.432	4.776	0.232	2.104
	2	0.614	5.733	0.247	2.138
	4	0.539	5.272	0.255	2.746

**Table 5 materials-16-05684-t005:** Results of ANOVA for the roughness parameter Ra.

Source	Sum of Squares	Degrees of Freedom	Mean Square	F-Value	*p*-Value	Meaning
Model	0.6451	2	0.3225	66.04	<0.0001	significant
A—drawing speed	0.0915	1	0.0915	18.72	0.0004	
B—measurement side	0.5536	1	0.5536	113.35	<0.0001	
Residual	0.0928	19	0.0049			
Total correlation	0.7379	21				

**Table 6 materials-16-05684-t006:** Fit statistics of the regression model for the roughness parameter Ra.

**Standard deviation**	0.0699	**R^2^**	0.8742
**Mean**	0.3895	**Adjusted R^2^**	0.8610
**Coefficient of variation. %**	17.94	**Predicted R^2^**	0.8178
		**Adequacy precision**	21.2885

**Table 7 materials-16-05684-t007:** Results of ANOVA for the the roughness parameter Rz.

Source	Sum of Squares	Degrees of Freedom	Mean Square	F-Value	*p*-Value	Meaning
Model	54.52	5	10.90	69.94	<0.0001	significant
A—angle of floating plug	0.5763	1	0.5763	3.70	0.0725	
B—drawing speed	0.0229	1	0.0229	0.1472	0.7063	
C—measurement side	48.04	1	48.04	308.14	<0.0001	
AB	0.9475	1	0.9475	6.08	0.0254	
B^2^	1.22	1	1.22	7.82	0.0129	
Residual	2.49	16	0.1559			
Total correlation	57.01	21				

**Table 8 materials-16-05684-t008:** Fit statistics of the regression model. for the roughness parameter Rz.

**Standard deviation**	0.3948	**R^2^**	0.9562
**Mean**	3.88	**Adjusted R^2^**	0.9426
**Coefficient of variation. %**	10.19	**Predicted R^2^**	0.9274
		**Adequacy precision**	22.6581

**Table 9 materials-16-05684-t009:** Results of ANOVA for the hardness.

Source	Sum of Squares	Degrees of Freedom	Mean Square	F-Value	*p*-Value	Meaning
Model	7122.09	5	1424.42	12.93	<0.0001	significant
A—angle of floating plug	2.72	1	2.72	0.0247	0.8757	
B—drawing speed	495.39	1	495.39	4.50	0.0391	
C—distance from outer surface	19.66	1	19.66	0.1784	0.6746	
A^2^	447.17	1	447.17	4.06	0.0495	
C^2^	5494.81	1	5494.91	49.87	<0.0001	
Residual	5399.40	49	110.19			
Total correlation	12,521.49	54				

**Table 10 materials-16-05684-t010:** Fit statistics of the regression model.

Standard deviation	10.50	R^2^	0.5688
**Mean**	268.51	**Adjusted R^2^**	0.5248
**Coefficient of variation. %**	3.91	**Predicted R^2^**	0.4470
		**Adequacy precision**	12.8640

## Data Availability

The data presented in this study are available on request from the corresponding author.

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
