# Peer review of "Cold Drawing of AISI 321 Stainless Steel Thin-Walled Seamless Tubes on a Floating Plug"

_materials, 2023, doi:10.3390/ma16165684_

Round 1

Reviewer 1 Report

This manuscript investigated the influence of the geometry of dies and plugs, drawing velocity and lubricants on the temperature in the drawing process, the mechanical properties of the tubes drawn, and the microhardness and roughness of the inner and outer surface of the tubes was investigated. The use of experimental designs could have helped to better structure the manuscript since there are several factors being investigated and several output parameters. Authors must present in detail all measurements performed. Some indication of the repeatability of the various measurands considered should be presented and discussed. All measurement systems used must be specified. The manuscript must undergo an extensive review before being considered for publication.

Some comments

1. According to the International System of Units (SI) 9th edition 2019,

a) When the symbol % is used, a space separates the number and the symbol %.

b) The numerical value always precedes the unit and a space is always used to separate the unit from the number. Thus, the value of the quantity is the product of the number and the unit. The space between the number and the unit is regarded as a multiplication sign (just as a space between units implies multiplication). The only exceptions to this rule are for the unit symbols for degree, minute and second for plane angle, °, ′ and ″, respectively, for which no space is left between the numerical value and the unit symbol.

These comments are valid for the whole manuscript.

2. On Abstract, “Tubes with dimensions (D0 × g0) 19 × 1.2 mm and 18 × 1.2 mm were drawn to produce tubes with dimensions (Dk × gk) 16 × 1.06 mm on a drawbench with the same total elongation, while the diameter and wall thickness were changed.”

a) Specify which tube dimensions are shown. What does D0, g0, Dk and gk mean? Define them when they first appear in the text.

b) Is 19 × 1.2 mm correct? The symbol × represents a multiplication, the unit would be mm2. Chek!

These comments are valid for the whole manuscript.

3. Write the name of the roughness parameters Ra and Rz the first time they appear in the text.

a) What is the justification for evaluating only two roughness parameters that are amplitude parameters?

4. The introduction has three pages. Couldn't it be smaller?

5. In the title of Figure 1, last line, there are two parentheses without text. Check!

6. On Page 4, Line 167 “The final outer diameter of tubes was Dk = 16 mm and wall thickness gk = 1.06 mm, respectively.”

a) How were measured the final outer diameter of tubes and the wall thickness? Were exact values obtained?

7. On Page 5, Lines 194 to 203 several measurement systems were cited.

a) Specify metrologically all measurement systems used. Declare the resolution and nominal range.

b) Specify whether these measurement systems were calibrated prior to measurements. This information is important to demonstrate that the obtained results are valid and traceable.

c) Specify how the measurements were taken. How many measurement cycles were performed in each case. This information is important for potential readers.

d) State the area that was considered during the topography obtaining. Specify whether any filters were used to obtain the effective topography.

d) Specify which cut-off was used during roughness measurements. Whether any other filters were applied on the raw profile.

8. Tables 1 and 3 are divided into two pages. Check!

9. „bamboo effect” Is this correct? Shouldn't it be "bamboo effect”?

This comment is valid for the whole manuscript.

10. On page 11, Line 350. State what FDP means.

11. On page 13, Line 379 “Microhardness in the middle part of the wall thickness was approx. 160HV, while”

a) What does approx. mean? Check the whole manuscript!

12. On Page 13, Linha 393, o nome do parãmetro de rugosidade Rz está errado. Veja ISO 4287. Geometrical Product Specifications (GPS) — Surface texture: Profile method — Terms, definitions and surface texture parameters. 2021.

13. The number of significant digits in the results shown in Table 4 must be check.

14. All results were shown as a single value. Authors must add error bars in the graphs indicating at least the standard deviation of the presented values.

This comment is also valid for the tables that present measurement results.

15. Several conditions were investigated. No statistical analysis was presented. The application of analysis of variance is recommended to analyze research results.

16. The manuscript is very extensive. The introduction and results could be rewritten in order to reduce the text.

Reviewer 2 Report

Dear Authors,
I have read your paper "Cold drawing of AISI 321 Stainless Steel Thin-Walled Seamless Tubes on a Floating Plug" carefully.
The paper is easy to read.
The paper is interesting. However, it requires few corrections.

1. Please, correct the abstract.
2. Please, discuss black spot on the surface of the 321 steel (fig.2 and table 3). Why is the surface clean after treatment?

3. Complete the conclusions with the limitations of the proposed methodology. Also write future research.

The paper can be accepted for publication after minor improvements.

Round 2

Reviewer 1 Report

Authors addressed almost all of my comments. The quality of the manuscript was improved, however some inaccuracies in the use of metrological parameters still remain. Thus, my decision is Minor Revision.

Minor concerns

a. How were measured the final outer diameter of tubes and the wall thickness? Were exact values obtained?

Author response: The outer diameter of the pipes was measured using an electronic caliper. The wall thickness was measured using an electronic micrometer.

In addition, for comparison and verification, outer diameter of tubes and the wall thickness was measured using the Atos Core 200 optical 3D scanner from GOM company. For this purpose, the pipes were cut into sections for testing mechanical properties and subjected to measurement of geometry as well as roughness and microhardness.

Reviewer: Specify the resolution and nominal range of the caliper and micrometer. State how many measurements were taken. Whether or not these measurement systems were calibrated prior to measurement. Also declare the ambient temperature at which the measurements were carried out with these two measurement systems.

7. On Page 5, Lines 194 to 203 several measurement systems were cited.

a) Specify metrologically all measurement systems used. Declare the resolution and nominal range.

Author response:

Measuring accuracy

- drawing force: 0,1 N

- thermocouple: ±1-2 °C,

- electronic caliper: 0.01 mm,

- electronic micrometer: 0.01 mm,

- optical 3D scanner: 0.02 mm

- testing machine: positioning accuracy and repeatability ± 2 μm/± 2 μm, force measurement - class 0.5 0.5/1

- microhardness tester: effective measurement range 250 μm, resolution 0.01 μm

- topography and roughness device: is fully automatic in a simple, bottom-table arrangement for observation in reflected light. Microscope uses UV laser light with a wavelength of 405 nm, resolution 10 nm

Reviewer: The resolution of all measurement systems must be declared. The measurement system resolution determines the number of significant digits that are shown in the measurement result. Accuracy and resolution are two different metrological terms. See the JCGM 200:2012 International vocabulary of metrology.

d) Specify which cut-off was used during roughness measurements. Whether any other filters were applied on the raw profile.

Reviewer: The authors did not specify the cut-off value. Other filters could also have been used, such as the Gauss filter.

On the surface of the part there is an overlap of form deviations such as flatness and cylindricity, for example, with waviness and roughness. The roughness signal has a lower amplitude and a higher frequency than the signals of form deviations and waviness. For this reason, you have to apply filters to be able to measure the roughness.

In ISO 4288 there is a table with cut-off values.

13. The number of significant digits in the results shown in Table 4 must be check.

Author response: The results of measurements of the Ra and Rz roughness parameters presented in Table 4 are presented with an accuracy of 0.01 μm.

Reviewer: It is not the accuracy that defines the number of significant figures. It's the resolution. Check!
